# Indirect impacts of the COVID-19 pandemic at two tertiary neonatal units in Zimbabwe and Malawi: an interrupted time series analysis

Simbarashe Chimhuya,[1] Samuel R Neal ⬩,[2] Gwen Chimhini,[1] Hannah Gannon,[2] Mario Cortina Borja,[2] Caroline Crehan,[2] Deliwe Nkhoma,[3] Tarisai Chiyaka,[4] Emma Wilson ⬩,[2] Tim Hull-Bailey,[2] Felicity Fitzgerald ⬩,[2] Msandeni Chiume,[5] Michelle Heys[2]

SC and SRN are joint first authors.
MC and MH are joint last authors.

For numbered affiliations see end of article.

**Correspondence to**
Dr Michelle Heys;
m.heys@ucl.ac.uk

## ABSTRACT

**Objectives** To examine indirect impacts of the COVID-19 pandemic on neonatal care in low-income and middle-income countries.

**Design** Interrupted time series analysis.

**Setting** Two tertiary neonatal units in Harare, Zimbabwe and Lilongwe, Malawi.

**Participants** We included a total of 6800 neonates who were admitted to either neonatal unit from 1 June 2019 to 25 September 2020 (Zimbabwe: 3450; Malawi: 3350). We applied no specific exclusion criteria.

**Interventions** The first cases of COVID-19 in each country (Zimbabwe: 20 March 2020; Malawi: 3 April 2020).

**Primary outcome measures** Changes in the number of admissions, gestational age and birth weight, source of admission referrals, prevalence of neonatal encephalopathy, and overall mortality before and after the first cases of COVID-19.

**Results** Admission numbers in Zimbabwe did not initially change after the first case of COVID-19 but fell by 48% during a nurses' strike (relative risk (RR) 0.52, 95% CI 0.41 to 0.66, p<0.001). In Malawi, admissions dropped by 42% soon after the first case of COVID-19 (RR 0.58, 95% CI 0.48 to 0.70, p<0.001). In Malawi, gestational age and birth weight decreased slightly by around 1 week (beta −1.4, 95% CI −1.62 to −0.65, p<0.001) and 300 g (beta −299.9, 95% CI −412.3 to −187.5, p<0.001) and outside referrals dropped by 28% (RR 0.72, 95% CI 0.61 to 0.85, p<0.001). No changes in these outcomes were found in Zimbabwe and no significant changes in the prevalence of neonatal encephalopathy or mortality were found at either site (p>0.05).

**Conclusions** The indirect impacts of COVID-19 are context-specific. While our study provides vital evidence to inform health providers and policy-makers, national data are required to ascertain the true impacts of the pandemic on newborn health.

## INTRODUCTION

The WHO declared COVID-19 a Public Health Emergency of International Concern on 30 January 2020.[1] Almost 2 years later,

## STRENGTHS AND LIMITATIONS OF THIS STUDY

⇒ We address the need for increased research into the indirect impacts of the COVID-19 pandemic on neonatal care in low-income and middle-income countries.
⇒ We collected data digitally and in real time using the Neotree application, which enabled a large sample size of 6800 neonates with minimal missing data.
⇒ It is possible that unobserved events occurred close to the first case of COVID-19 in either country, which could have influenced our results.
⇒ We only collected data on neonates admitted to the neonatal unit and did not capture stillbirths or neonatal deaths that occurred in the community.

confirmed cases have exceeded 281 million globally with over 5.4 million deaths to the end of 2021.[2] Zimbabwe recorded its first case on 20 March 2020 and, to date, has reported over 200 000 cases with nearly 5000 deaths.[2] Malawi confirmed its first three cases on 3 April 2020 and has reported more than 72 000 cases and over 2000 deaths in this same period.[2]

Before the COVID-19 pandemic, considerable improvements were made in global child health: the global neonatal mortality rate fell from 31 to 18 deaths per 1000 live births between 2000 and 2018.[3] Yet there were disparities in the rates of decline with the sub-Saharan Africa region facing highest neonatal mortality rates.[3] Now, there is a danger that health outcomes in low-income and middle-income countries (LMICs) will fall further behind high-income countries. While countries worldwide face challenges related to the COVID-19 pandemic, LMICs are particularly struggling with financial constraints, limited testing capacity, lack of personal protective equipment, staff shortages,[4 5] and limited

access to vaccines.[6] As children are at low risk of infection or severe disease from COVID-19,[7–11] any impacts on their health outcomes will likely be attributable to the indirect effects of the pandemic on health systems, as in previous disease outbreaks.[12 13] These include increased rates of parental unemployment, food and housing insecurity, and reduced access to routine care, including antenatal and perinatal care, with potentially damaging downstream impacts on neonatal outcomes.[14 15]

We hypothesised that the COVID-19 pandemic would negatively impact care seeking behaviours, neonatal care provision and, ultimately, neonatal outcomes in LMICs. To test this hypothesis, we aimed to examine trends in markers of neonatal care before and during the initial months of the COVID-19 pandemic at Sally Mugabe Central Hospital (SMCH), Zimbabwe and Kamuzu Central Hospital (KCH), Malawi. Specifically, we compared the:
1. Number of admissions to the neonatal unit (NNU).
2. Gestational age and birth weight of admitted neonates.
3. Source of admission referrals.
4. Prevalence of neonatal encephalopathy (NE).
5. Overall mortality rate before and after the first reported cases of COVID-19.

## METHODS

This study is reported in accordance with the Strengthening the Reporting of Observational Studies in Epidemiology statement (online supplemental appendix 1).

### Setting

#### Health facilities

SMCH is a public referral hospital in Harare, Zimbabwe. It has the largest of three tertiary NNUs nationwide with 100 cots. KCH, Lilongwe, is one of four regional referral hospitals in Malawi and the NNU has 75 cots. Neonatal care at SMCH is predominantly doctor led while neonatal care at KCH is mostly nurse led. Both units accept local and national referrals for specialist surgical care.

#### Government response to the pandemic

In response to the COVID-19 pandemic, Zimbabwe and Malawi both implemented response measures in an attempt to control the outbreak. In Zimbabwe, the Government closed borders to non-essential travel within days of the first in-country confirmed case of COVID-19 and imposed a full national lockdown that lasted from 30 March to 11 June 2020, which was followed by phased relaxations of the restrictions.[16] In Malawi, public events were banned and public gatherings restricted to fewer than 100 people on 20 March 2020, with all educational institutions closed several days later.[17] Borders were closed to non-essential travel on 1 April 2020 and a full national lockdown was announced to last for 21 days from 18 April 2020; however, a High Court injunction prevented this. Further restrictions were announced on 9 August 2020, mandating the wearing of face masks in public, closing places of worship, restaurants, and bars, and restricting public gatherings to less than 10 people initially, although

these were revised within days to reallow gatherings up to 100 people.[18]

### Industrial action by health workers in Zimbabwe

Two periods of national industrial action occurred in Zimbabwe during our study. Doctors went on strike from 3 September 2019 to 22 January 2020 (pre-COVID-19 period) citing insufficient pay and poor working conditions, which put significant pressure on the public health system.[19] Additionally, there was a period of strikes by nurses from 17 June to 9 September 2020 (post-COVID-19 period) over pay and availability of personal protective equipment during the pandemic.[20]

### Participants

All neonates admitted to each NNU over a 16-month period from 1 June 2019 to 25 September 2020 (69 complete weeks) were eligible for inclusion. We applied no specific exclusion criteria.

### Data collection

Data were collected prospectively using the Neotree application (app), an Android tablet-based quality improvement platform that aims to reduce neonatal mortality in low-resource settings.[21] Developed in collaboration with local stakeholders, it is embedded in routine practice at two NNUs in Zimbabwe and Malawi, providing real-time clinical decision support, neonatal care education and digital data capture.[22 23]

Health workers complete a digital form when a neonate is admitted to the unit (admission form) and when they are discharged or die (outcome form). The app guides assessment of the neonate and collects data on patient demographics, examination findings, diagnoses and interventions. Pseudonymised forms are uploaded monthly to University College London servers (Zimbabwe data) and Amazon Web Services (Malawi data). Admission and outcome forms are linked by a unique identifier generated by the app at admission.

### Outcomes

We evaluated five outcomes:
1. Number of admissions: determined from the admission date of each completed admission form.
2. Gestational age at birth (weeks) and birth weight (grams): as entered into the admission form from obstetric records.
3. Source of admission: defined as 'within' (labour ward, postnatal ward, antenatal ward, obstetric theatre or fee-paying ward (KCH only)) or 'outside' (referral from another health facility or postnatal self-referral from home).
4. Diagnosis of NE: defined as 'hypoxic ischaemic encephalopathy' or 'birth asphyxia' recorded as a diagnosis, cause of death or contributory cause of death on the outcome form.
5. Mortality: defined as an outcome of "neonatal death" on the outcome form. All other neonates, including

those discharged, transferred to another facility or who left on parental request, were considered alive.

## Statistical analysis

Analyses were performed in R V.3.6.3,[24] running on RStudio V.1.2.5033.[25] First, admission forms were matched with their corresponding outcome form based on the unique identifier generated at admission. Lack of completed outcome forms (SMCH: n=325 (9.4% of admission forms completed); KCH: n=245 (7.3%)) or errors in entry of the unique identifier at discharge (SMCH: n=310 (9.9% of outcome forms completed); KCH: n=182 (5.9%)) meant we were unable to match some admission forms with outcome forms (SMCH: n=635 (18.4% of admission forms completed); KCH: n=427 (12.7%)). For outcomes 1–3, we based analyses on data from all admission forms, regardless of match status. For outcomes 4 and 5, we based analyses on matched records only. Matched records implying a negative admission duration (ie, outcome date prior to admission date) were excluded (SMCH: n=57 (2.0% of matched records); KCH: n=24 (0.8%)). See online supplemental appendix 3 for a flow diagram of record inclusion. Missing data were excluded using pairwise deletion for each analysis as frequencies of missing values were minimal (online supplemental appendix 4).

This study used an interrupted time series design with weekly data windows. We considered the first confirmed case of COVID-19 in each country as the intervention (Zimbabwe: 20 March 2020; Malawi: 3 April 2020).[2] For all outcomes, we hypothesised a level change impact model without a lag, and this was tested using interrupted time series regression models.[26] Gestational age and birth weight were modelled with linear regression. Count data were modelled using generalised linear models with Poisson or negative binomial responses and logarithmic link functions. We assessed for dispersion by dividing the residual deviance by the df for the Poisson model. Where this quotient was much greater than one (greater than approximately 1.10) we instead used a negative binomial model to account for overdispersion. Accordingly, source of admission referral, prevalence of NE and overall mortality at SMCH were modelled using Poisson models, while number of admissions and overall mortality at KCH were modelled using negative binomial models.

All models for SMCH were adjusted for the periods of doctors' strikes (3 September 2019 to 22 January 2020) and nurses' strikes (17 June to 9 September 2020). For count data, we adjusted for variation in the number of admissions over time by including the logarithm of the number of admissions in each weekly window as an offset term. Presence of autocorrelation was assessed using autocorrelation function plots and by examining models' residuals. Seasonality was included in the interrupted time series models with cosine functions with variable amplitude and shift. We tested models fitting cosine functions on week of admission with 6-month and 12-month periods, and a model including these two harmonic terms.

To achieve this, we transformed each cosine function into a sine term and cosine term, and included these terms in the regression models for each outcome (as described by Stolwijk et al).[27] The final models presented were selected by minimising the Bayesian information criterion and by comparing goodness-of-fit with the $\chi^2$ test for nested models. Adjusting for seasonality did not improve the fit of any of the models tested and, thus, all presented models are unadjusted for seasonality. See online supplemental appendix 5 for model selection and estimates.

## Patient and public involvement

Although patients and the public were not directly involved in this study, within the broader Neotree co-development project we are carrying out a series of workshops and focus group discussions with healthcare workers and parents of admitted babies to ensure local ownership and relevance of this digital quality involvement tool aimed at improving healthcare outcomes for vulnerable neonates.

## RESULTS

### Outcome 1: admissions to the NNU

We included 3450 neonates at SMCH and 3350 neonates at KCH. Figure 1 shows the 7-day moving average of admissions to the NNU.

At SMCH, the mean (SD) number of weekly admissions was 54.6 (23.5) before the first case of COVID-19 (pre-COVID-19) and 42.8 (19.9) afterwards (post-COVID-19). The negative binomial regression model showed no evidence of a change in admissions after the first case of COVID-19 (relative risk (RR) 0.87; 95% CI 0.65 to 1.17; p=0.37) (figure 2A). However, this model estimated that admissions fell by 48% during the nurses' strike period (RR 0.52, 95% CI 0.41 to 0.66; p<0.001) and by 51% during the pre-COVID-19 doctors' strikes (RR 0.49, 95% CI 0.41 to 0.60; p<0.001).

At KCH, the mean (SD) number of weekly admissions was 54.5 (10.8) in the pre-COVID-19 period and 38.0 (10.9) in the post-COVID-19 period. The negative binomial regression model yielded a 42% reduction in admissions after the first case of COVID-19 (RR 0.58; 95% CI 0.48 to 0.70; p<0.001) (figure 2B).

### Outcome 2: gestational age and birth weight

At SMCH, the mean (SD) gestational age at birth was 36.1 (4.4) weeks in the pre-COVID-19 period and 36.0 (4.2) weeks in the post-COVID-19 period. The mean (SD) birth weight was 2500 (908) g in the pre-COVID-19 period and 2487 (896) g in the post-COVID-19 period. Linear regression analysis indicated no significant change in gestational age at birth nor birth weight after the first case of COVID-19 (gestational age: beta 0.07; 95% CI −0.50 to 0.64; p=0.81, birth weight: beta 3.4; 95% CI −117.0 to 123.8; p=0.96) (online supplemental figure 1A, C).

At KCH, the mean (SD) gestational age was 35.0 (3.9) weeks in the pre-COVID-19 period and 34.8 (3.9) weeks in the post-COVID-19 period. The mean (SD) birth weight

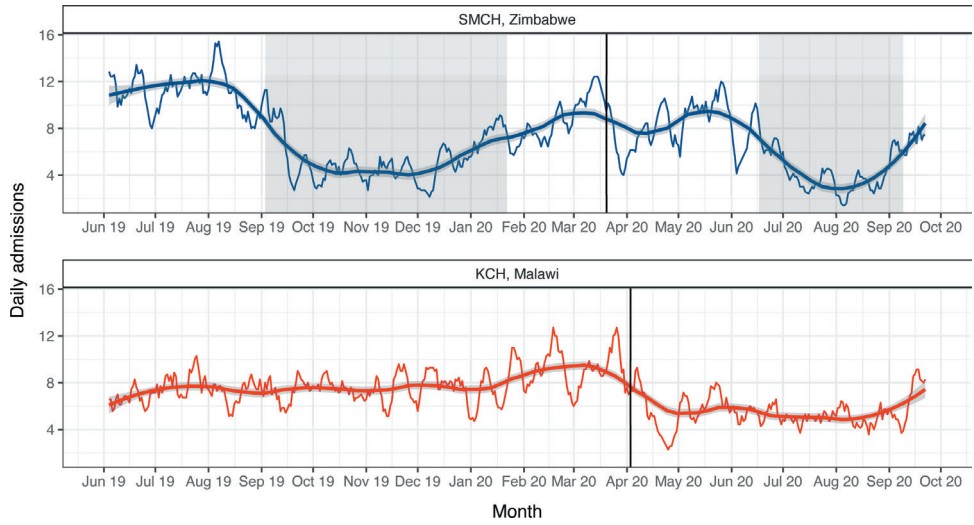

**Figure 1** Trend in daily admissions to the neonatal unit. The 7-day moving average of daily admission numbers has been plotted. Smoothed line: local regression (LOESS) model fitted on the 7-day moving average of daily admission numbers; shaded region: 95% CI. Solid vertical line: first confirmed case of COVID-19 in each country. Shaded periods on SMCH, Zimbabwe panel: industrial action by doctors (3 September 2019 to 22 January 2020) and nurses (17 July 2020 to 9 September 2020). Counts based on all admission forms completed, irrespective of match status. KCH, Kamuzu Central Hospital; SMCH, Sally Mugabe Central Hospital.

was 2402 (883) g in the pre-COVID-19 period and 2299 (870) g in the post-COVID-19 period. Gestational age significantly decreased by 1 week in the post-COVID-19 period (beta −1.14; 95% CI −1.62 to −0.65; p<0.001) (online supplemental figure 1B) and birth weight significantly decreased by 300 g (beta −299.9; 95% CI −412.3 to −187.5; p<0.001) (online supplemental figure 1D).

### Outcome 3: source of admission referral

At SMCH, the mean (SD) percentage of outside referrals to the NNU was 39 (11)% in the pre-COVID-19 period and 35 (9)% in the post-COVID-19 period. The Poisson regression model showed no evidence of a change in

the percentage of outside referrals after the first case of COVID-19 (RR 0.97; 95% CI 0.77 to 1.22; p=0.81) (figure 3A). However, this model did imply a 39% relative increase in the percentage of outside referrals during the doctors' strikes in the pre-COVID-19 period (RR 1.39; 95% CI 1.20 to 1.61; p<0.001).

At KCH, the mean (SD) percentage of outside referrals was 61 (8)% in the pre-COVID-19 period and 51 (10)% in the post-COVID-19 period. Poisson regression analysis resulted in a 28% relative reduction in outside referrals after the first case of COVID-19 (RR 0.72; 95% CI 0.61 to 0.85; p<0.001) (figure 3B).

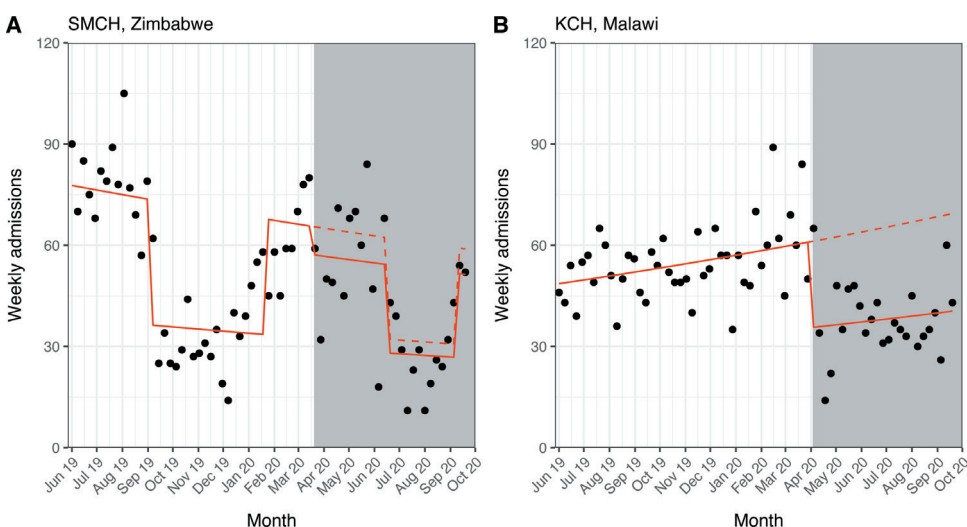

**Figure 2** Interrupted time series for weekly admissions to the neonatal unit. White background: pre-COVID-19 period; grey background: post-COVID-19 period. Solid line: predicted trend from negative binomial regression model; dashed line: counterfactual scenario. SMCH model (A) adjusted for doctors' and nurses' strike periods; KCH model (B) unadjusted. Counts based on all admission forms completed, irrespective of match status. KCH, Kamuzu Central Hospital; SMCH, Sally Mugabe Central Hospital.

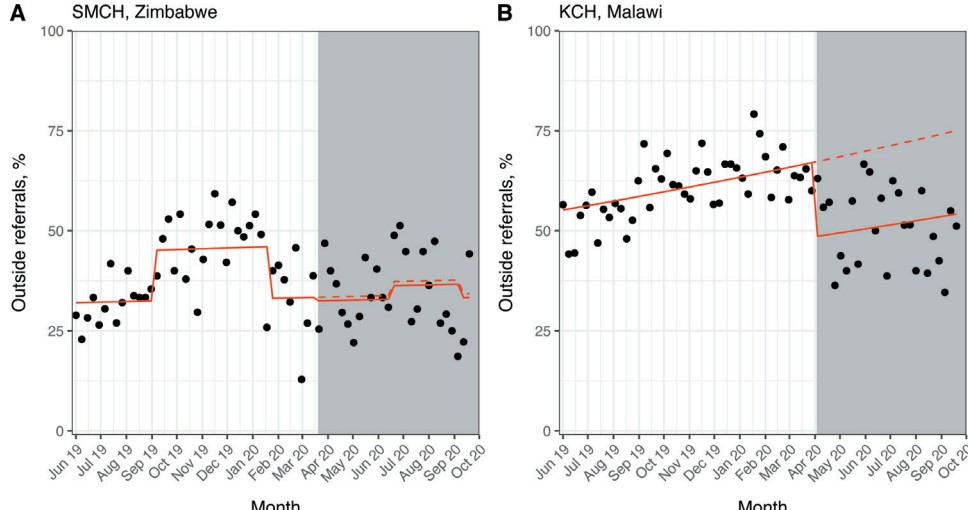

**Figure 3** Interrupted time series for outside referrals to the neonatal unit. White background: pre-COVID-19 period; grey background: post-COVID-19 period. Solid line: predicted trend from Poisson regression model; dashed line: counterfactual scenario. SMCH model (A) adjusted for doctors' and nurses' strike periods, KCH model (B) unadjusted. Data from all admission forms completed, irrespective of match status. KCH, Kamuzu Central Hospital; SMCH, Sally Mugabe Central Hospital.

## Outcome 4: prevalence of NE

At SMCH, the mean (SD) percentage of admitted neonates diagnosed with NE was 16 (6)% in the pre-COVID-19 period and 21 (12)% in the post-COVID-19 period suggesting a possible increase. Poisson regression analysis showed no statistically significant change in the percentage of neonates diagnosed with NE post-COVID-19 (RR 1.06; 95% CI 0.74 to 1.52; p=0.74) (online supplemental figure 2A).

At KCH, the mean (SD) percentage of admitted neonates diagnosed with NE was 15 (6)% in the pre-COVID-19 period and 13 (5)% in the post-COVID-19 period. The Poisson regression model implied a possible increase in diagnoses of NE after the first case of COVID-19, but this

was not statistically significant (RR 1.31; 95% CI 0.91 to 1.88; p=0.15) (online supplemental figure 2B).

## Outcome 5: overall mortality

For SMCH, the mean (SD) percentage of deaths per week of admission was 25 (10)% in the pre-COVID-19 period and 26 (16)% in the post-COVID-19 period. The negative binomial regression model pointed towards a possible decrease in mortality after the first case of COVID-19, but this was not statistically significant (RR 0.72; 95% CI 0.52 to 1.00; p=0.05) (figure 4A). However, this model did show an 81% relative increase in mortality during the nurses' strike period (RR 1.81; 95% CI 1.31 to 2.49; p<0.001).

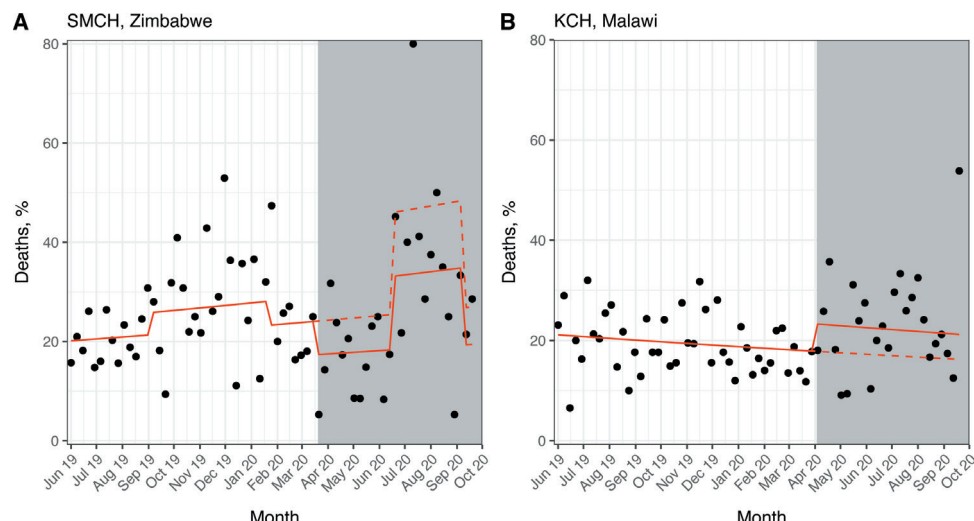

**Figure 4** Interrupted time series for overall mortality. White background: pre-COVID-19 period; grey background: post-COVID-19 period. Solid line: predicted trend from negative binomial regression model (SMCH, A) or Poisson regression model (KCH, B); dashed line: counterfactual scenario. SMCH model (A) adjusted for doctors' and nurses' strike periods; KCH model (B) unadjusted. Data from matched admission and outcome forms only. KCH, Kamuzu Central Hospital; SMCH, Sally Mugabe Central Hospital.

For KCH, the mean (SD) percentage of deaths per week of admission was 19 (6)% in the pre-COVID-19 period and 23 (10)% in the post-COVID-19 period. The Poisson regression model implied a possible increase in mortality after the first case of COVID-19, but this was not statistically significant (RR 1.31; 95% CI 0.97 to 1.76; p=0.08) (figure 4B).

## DISCUSSION
### Summary

We performed an interrupted time series analysis to examine changes in neonatal care provision at two tertiary NNUs in Zimbabwe and Malawi after the first cases of COVID-19 in each country. We found that admissions at SMCH did not change significantly after the first case of COVID-19 when considering this period as a whole, but there was a considerable decrease (around 50%) in the number admissions in June to August 2020, coinciding with a nurses' strike. We did not find significant changes in gestational age or birth weight, source of admission referrals, prevalence of NE or mortality at SMCH. Conversely, we found several changes in markers of neonatal care at KCH after the first case of COVID-19 in Malawi. The number of admissions fell by 42% and we noted a decrease in the gestational age and birth weight of admitted neonates (by around 1 week and 300 g, respectively), and a 28% relative decrease in outside referrals after the first case of COVID-19. Although this study is descriptive, we can speculate about explanations for our results based on existing literature and discussions with local health workers.

### Interpretation

The number of admissions at SMCH fell by around 50% between June to August 2020, but we noted no change

outside this strike period, suggesting some resilience to the impact of the pandemic. However, nurses went on strike over pay and availability of personal protective equipment,[20] so the strike is itself an indirect consequence of COVID-19. A recently published audit of maternal health service provision at two tertiary hospitals in Harare, Zimbabwe (including SMCH) found a 25% reduction in hospital deliveries and an increased odds of stillbirth (OR 1.8; 95% CI 1.5 to 2.2) in March to August 2020 compared with the same period in 2019,[28] which might partially explain the reduction in admissions to the NNU. A similar reduction in admissions was seen at KCH, but, unlike at SMCH, this 42% decrease was noted within a week of the first case of COVID-19. In figure 5, we propose several interlinked factors that might explain reduced admissions to the NNU. Several of these factors, such as fear of using health services, disrupted transport networks and staff shortages have been directly reported by local sources in low-resource settings and were highlighted in a recent report by Graham *et al.*[29]

We found a slight decrease in gestational age and birth weight of neonates at KCH, but not SMCH. Studies have reported increased rates of preterm birth in pregnant women with COVID-19 compared with those without the disease, mostly from medically induced preterm birth; although none of these studies were conducted in LMICs.[30] Preliminary analysis suggests rates of emergency caesarean section increased at SMCH and KCH, with a more marked increase at KCH (online supplemental appendix 6). This is one potential explanation for our findings. However, we noted that the number of outside referrals decreased by 28% at KCH, and neonates referred from outside KCH are more likely to be from lower-risk pregnancies that delivered in a health centre with higher gestational ages and birth weights. Further

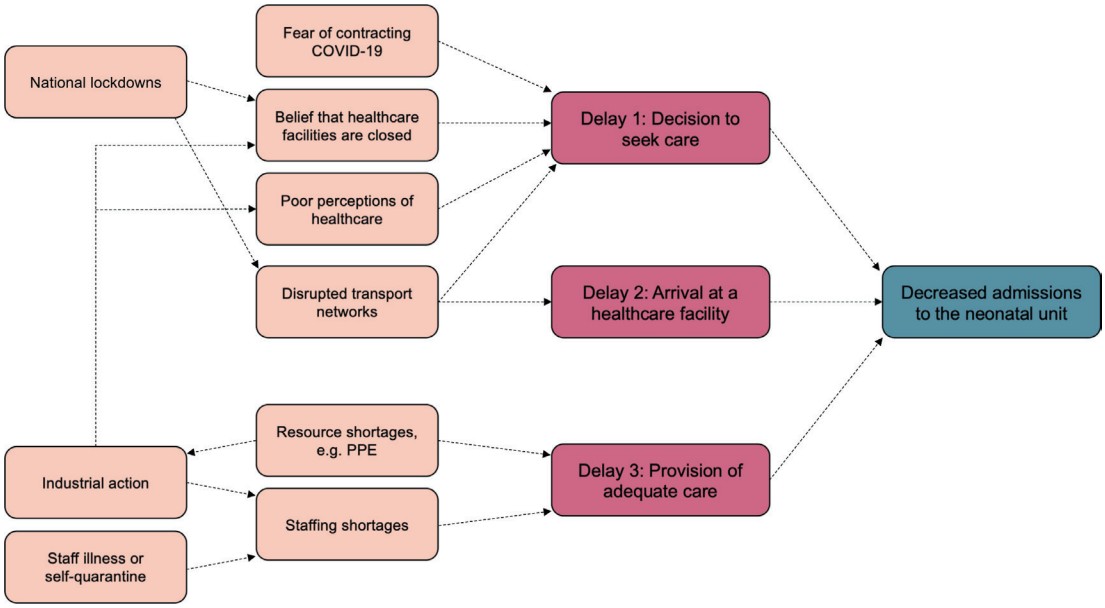

**Figure 5** Possible factors influencing the decrease in admissions to the neonatal unit. Delays (red boxes) derived from the 'three delays' model of pregnancy-related mortality.[36] PPE, personal protective equipment.

analysis should stratify by source of admission referral to clarify this finding, but the relative reduction in outside referrals is supported by the fact that referrals were rigorously triaged by the on-call paediatrician during the pandemic, and that referrals from some areas were diverted away from KCH to more appropriate centres for the level of care required.

We hypothesised that rates of NE would increase during the pandemic. NE is the clinical manifestation of disordered brain function and can have multiple aetiologies.[31] The term 'hypoxic-ischaemic encephalopathy' is reserved for cases where there is evidence of intrapartum asphyxia.[31] In LMICs, obstructed labour is a major cause of maternal mortality and can lead to intrapartum asphyxia with subsequent neonatal morbidity and mortality, including NE.[32] Therefore, the prevalence of NE might be expected to increase as a marker of delayed presentation to a health facility. It is reassuring that we did not find increased rates of NE at SMCH or KCH. However, these findings should be interpreted cautiously as some neonates with NE may not have presented to a health facility at all, for example, due to an increased number of home deliveries, as documented in other sub-Saharan countries.[33]

Finally, we did not observe a significant change in overall mortality at KCH nor SMCH, except during the nurses' strikes at SMCH. In fact, there was a suggestion that mortality decreased after the first case of COVID-19 in Zimbabwe when adjusted for the nurses' strike period, but this was not statistically significant. The reasons for this are unclear but could include factors such as increased stillbirth rates or improved care for the smaller number of neonates on the NNU. More complete analysis of facility-based and community-based neonatal mortality is greatly needed.

## Limitations and future work

A limitation intrinsic to interrupted time series analysis is the possibility that another event occurred close to the first case of COVID-19 in either country causing spurious observations. Another potential threat to validity is changing data collection practices. For example, overstretched clinicians might not input data into the Neotree app for all admitted neonates. However, this is unlikely as the Neotree app is embedded into routine practice at SMCH and KCH and discussions with local collaborators suggest use of the app has continued without issue. At present, there is limited guidance on power and sample size calculations for interrupted time series analyses.[34] Therefore, we did not perform specific power calculations and relied on the data available at the time of analysis. Also, our results suggest that our study has relatively low power to detect true changes in some outcomes, particularly NE, so these results should be interpreted cautiously in the absence of further data.

The Neotree app only collects data on neonates admitted to the NNU. Therefore, our analysis does not capture stillbirths or neonatal deaths that occur in the community. It is troubling to see a dramatic fall in admissions at both sites, raising the possibility that many unwell neonates did not attend a health facility and died at home. A recent study found that facility births decreased by over 50% during the lockdown in Nepal, and facility stillbirth and neonatal mortality rates increased significantly.[35] The Neotree research team is currently collecting data on stillbirths at SMCH and KCH, but these data will still only represent stillbirths that occurred in a health facility. Given the COVID-19 pandemic is not over, it will be important to repeat our analysis to further examine longer-term trends in neonatal care provision.

## CONCLUSION

The indirect impacts of COVID-19 are context-specific, with more significant and evident effects on neonatal care provision seen at KCH (Malawi) than SMCH (Zimbabwe). While this study provides vital evidence to inform health providers and policy makers, national data are required to ascertain the true impacts of the pandemic on newborn health.

**Author affiliations**
[1]Child and Adolescent Health Unit, University of Zimbabwe, Harare, Zimbabwe
[2]UCL Great Ormond Street Institute of Child Health, University College London, London, UK
[3]Parent and Child Health Initiative, Lilongwe, Malawi
[4]Biomedical Research and Training Institute, Harare, Zimbabwe
[5]Department of Paediatrics, Kamuzu Central Hospital, Lilongwe, Malawi

**Acknowledgements** We are very grateful to the families at SMCH and KCH, and the staff members at both hospitals for their enthusiasm and commitment to the Neotree project, without which this work would not be possible.

**Contributors** Concept and study design by SC, SRN, GC, FF, MCB, CC, MC and MH with input from other authors. Data collected by HG, DN, TC, CC and TH-B. Analysis performed by SRN and MCB with contributions from FF, SC, EW and MH. Manuscript drafted by SC and SRN with input from GC, FF, MCB, MC and MH. All authors proofread and approved final draft. Underlying data accessed and verified by SRN, MCB, HG, FF and MH. MH is the guarantor of the study.

**Funding** We would like to thank the funders of this study. SRN was awarded the International Child Health Group David Morley Elective Bursary for this elective project. Funders of the wider Neotree project, past and present, include the Wellcome Trust Digital Innovation Award (215742/Z/19/Z: PI: Heys), RCPCH, Naughton-Cliffe Mathews, UCL Grand Challenges and Global Engagement Fund, and the Healthcare Infection Society (SRG 201802004). FF is supported by the Academy of Medical Sciences and the funders of the Starter Grants for Clinical Lecturers scheme. This study and MH and FF are further supported by the National Institute for Health Research Great Ormond Street Hospital Biomedical Research Centre. The funders had no role in study design, data collection and analysis, or preparation of this report.

**Disclaimer** The views expressed are those of the authors and not necessarily those of the National Health Service (NHS), the NIHR or the UK Department of Health. The funders had no role in study design, data collection, data analysis, data interpretation, or preparation of this manuscript.

**Competing interests** None declared.

**Patient and public involvement** Patients and/or the public were involved in the design, or conduct, or reporting, or dissemination plans of this research. Refer to the Methods section for further details.

**Patient consent for publication** Not applicable.

**Ethics approval** Research ethics approval was granted by the UCL Research Ethics Committee (17123/001) and ethics committees in Malawi (P.01/20/2909) and Zimbabwe (MRCZ/A/2570) (online supplemental appendix 2). The need to

obtain informed consent was waived as we collected only pseudonymised data routinely documented for clinical care.

**Provenance and peer review** Not commissioned; externally peer reviewed.

**Data availability statement** Data are available on reasonable request. Data collected for the study cannot yet be made publicly available because primary analysis for the pilot implementation evaluation of the Neotree, as well as secondary analysis is ongoing. A goal of our pilot implementation is to establish an open-source anonymised research database of Neotree data to maximise the reach and utility for researchers aiming to improve outcomes for neonates in low-income settings. This database is under development and subject to negotiation with relevant Ministries of Health.

**ORCID iDs**
Samuel R Neal http://orcid.org/0000-0001-6832-9839
Emma Wilson http://orcid.org/0000-0001-7091-2417
Felicity Fitzgerald http://orcid.org/0000-0001-9594-3228

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
