## [Reviewer comments · BMJ Open]

ARTICLE DETAILS

TITLE (PROVISIONAL)	Indirect impacts of the COVID-19 pandemic at two tertiary neonatal units in Zimbabwe and Malawi: an interrupted time series analysis
AUTHORS	Chimhuya, Simbarashe; Neal, Samuel; Chimhini, Gwen; Gannon, Hannah; Cortina Borja, Mario; Crehan, Caroline; Nkhoma, Deliwe; Chiyaka, Tarisai; Wilson, Emma; Hull-Bailey, Tim; Fitzgerald, Felicity; Chiume, Msandeni; Heys, Michelle

VERSION 1 – REVIEW

REVIEWER	Griffin, Ian Biomedical Research Institute of New Jersey
REVIEW RETURNED	24-Feb-2021

GENERAL COMMENTS	My main concern with this paper was related to my confusion about what the authors were looking at; specifically, what their hypothesis(es) were. I think I found that in their interpretation section ("we hypothesized that rates of NE would increase during the pandemic"). This needs to be stated far earlier, and the introduction needs to relate to this hypothesis. ABSTRACT - I'm not sure that the first cases of COVID count as an intervention INTRODUCTION - Some statement of the hypotheses is needed. Is the suggestion that these changes are a result of COVID per se, or in caregiver behavior in response to COVID and a desire to avoid hospitals? METHODS - Seem appropriate RESULTS - The results are well structured and the use of subheadings improves readability - The data for NE is limited to relatively low power to detect a change DISCUSSION - Is appropriate and balanced
--

REVIEWER	Okomo, Uduak London School of Hygiene & Tropical Medicine, MRC Unit The Gambia
REVIEW RETURNED	06-Mar-2021

GENERAL COMMENTS	Thank you for the opportunity to review this manuscript investigating
---

the indirect impact of the COVID-19 pandemic in tertiary neonatal units in sub-Saharan Africa, a region already disproportionately burdened with highest neonatal mortality rates.
Please find my specific comments below:

Major Comments

Introduction:

1. The first paragraph reads:“Zimbabwe recorded its first case on 20 March and has reported >17000 cases with >400 deaths to date.2 Malawi confirmed its first three cases on 3 April and has reported >7000 cases and ~200 deaths to date. 2.....
The number of reported cases and deaths due to COVID-19 in both countries was current up to the date of manuscript submission and would therefore, need to be revised. Alternatively, the authors could give the exact date for which the data applies.

Methods

2. The setting section reads.... “Lilongwe, is one of four regional referral hospitals in Malawi and the NNU has 75 cots. In contrast to SMCH, care in the NNU is mostly nurse-led.”
This appears incomplete. Could the authors clarify if they are contrasting overall care at the SMCH with care at the SMCH NNU or that of KCH as well.

3. Although I am not a statistician, I have my reservations about the appropriateness of the choice of study design for this research question. The Interrupted time series analysis is a method of statistical analysis involving tracking a long-term period before and after a point of intervention to assess the intervention's effects. The time series refers to the data over the period, while the interruption is the intervention, which is a controlled external influence or set of influences. In this regard:

a. The date of first COVID-19 diagnosis, while a key event, cannot be considered as an intervention per se as it is not controlled. It is not the date of diagnosis but the government and health system response to this event (a lockdown) that is the ‘controlled intervention’ that has both direct and indirect impact. The authors failed to mention if there was any national or regional lockdown in either country or the duration of such a lockdown.

b. A bit more information on the setting in each NNU is necessary to provide context especially given the limitations intrinsic to interrupted time series analysis, one of which the author have mentioned to be the possibility that another event occurred close to the first case of COVID-19 in either country causing spurious observations. What is the usual pattern of admissions to each unit over the preceding few years? Is there usually any seasonal variation in deliveries at each hospital and NNU admissions? (In some studies, neonatal ward admissions have been shown to follow a seasonal pattern consistently peaking at certain months of the year. One needs at least 2 years data to establish seasonality). It is therefore important when considering any changes to the pattern of admissions that the authors compare events over the same times of the year before and after the intervention. If the first case of COVID-19 in either (or both) countries occurred about the same time that there is usually a drop in NNU admissions, then the authors cannot simply conclude their observations are due to the COVID-19 and would need to adjust for any seasonal trends. More appropriate would be the additional or excess decline in admissions above what would be expected for the same time of the year.

VERSION 1 – AUTHOR RESPONSE

Reviewer 1: Dr Ian Griffin

R1.1: “My main concern with this paper was related to my confusion about what the authors were looking at; specifically, what their hypothesis(es) were. I think I found that in their interpretation section (“we hypothesized that rates of NE would increase during the pandemic”). This needs to be stated far earlier, and the introduction needs to relate to this hypothesis.”

Thank you for highlighting this confusion. We intended this study to describe changes in several markers of neonatal care (specifically, the number of admissions to the neonatal unit, gestational age and birth weight, source of admission referrals, prevalence of neonatal encephalopathy, and overall mortality) in the first six months after the first case of COVID-19 in Malawi and Zimbabwe. There are limited data on the impacts of COVID-19 in low-income and middle-income countries, so we hoped this would partly address this literature gap.

To make our overall hypothesis clearer, we have now added the following to our ‘Introduction’ section (see also our response to point R1.3 below):

Page 6, line 47 onwards now reads (added text in blue italics):

These include increased rates of parental unemployment, food and housing insecurity, and reduced access to routine care, including antenatal and perinatal care, with potentially damaging downstream impacts on neonatal outcomes.^{13 14}

Page 7, line 10 onwards now reads (added text in blue italics):

We hypothesised that the COVID-19 pandemic would negatively impact care seeking behaviours, neonatal care provision and, ultimately, neonatal outcomes in LMICs. To test this hypothesis, we aimed to examine trends in markers of neonatal care...

Abstract

R1.2: “I’m not sure that the first cases of COVID count as an intervention”

Reviewer 2 also raised this concern. Please see our response to point R2.3 below where we address this comment in detail.

Introduction

R1.3: “Some statement of the hypotheses is needed. Is the suggestion that these changes are a result of COVID per se, or in caregiver behavior in response to COVID and a desire to avoid hospitals?”

We have now added the following statement of our overall hypothesis to the ‘Introduction’ section:

Page 7, line 10 onwards now reads (added text in blue italics):

We hypothesised that the COVID-19 pandemic would negatively impact care seeking behaviours, neonatal care provision and, ultimately, neonatal outcomes in LMICs. To test this hypothesis, we

aimed to examine trends in markers of neonatal care...

Methods

R1.4: "The data for NE is limited to relatively low power to detect a change"

We agree with Dr Griffin that our analysis for the change in prevalence of neonatal encephalopathy before and after the first cases of COVID-19 has relatively low power to detect a true effect. We hypothesised that the rates of neonatal encephalopathy would increase during the pandemic, and our point estimates of the effects with the available data match the direction of this hypothesis (RR 1.08; 95%CI 0.76-1.55; $p = 0.67$ for SMCH, and RR 1.30; 95%CI 0.95-1.80; $p = 0.11$ for KCH), but neither result was statistically significant.

We emphasise in the 'Results' section that there may have been an increase in diagnoses of neonatal encephalopathy after the first case of COVID-19 (particularly at KCH), but that this was not statistically significant at the 0.05 level. To clarify this, we have added the following to the 'Limitations and future work' section:

Page 18, line 29 onwards now reads (added text in blue italics):

At present, there is limited guidance on power and sample size calculations for interrupted time series analyses.²⁸ Therefore, we did not perform specific power calculations and relied on the data available at the time of analysis. Our results suggest that our study has relatively low power to detect true changes in some outcomes, particularly NE, so these results should be interpreted cautiously in the absence of further data.

Reviewer 2: Dr Uduak Okomo

Introduction

R2.1: "The first paragraph reads: 'Zimbabwe recorded its first case on 20 March and has reported >17000 cases with >400 deaths to date.² Malawi confirmed its first three cases on 3 April and has reported >7000 cases and ~200 deaths to date. 2.....'

The number of reported cases and deaths due to COVID-19 in both countries was current up to the date of manuscript submission and would therefore, need to be revised. Alternatively, the authors could give the exact date for which the data applies."

We have amended the first paragraph of the Introduction to provide the exact date to which the incidence data refers. We have chosen not to update this data so that the numbers of reported cases and deaths due to COVID-19 reflect the situation when our study was conducted, and the manuscript was initially written and submitted.

Page 6, lines 5-18 now read (added text in blue italics):

The World Health Organization declared coronavirus disease (COVID-19) a Public Health Emergency of International Concern on 30 January 2020.¹ As of 21 October 2020, confirmed cases have exceeded 80 million globally with nearly 2 million deaths.² Zimbabwe recorded its first case on 20 March and, up to 21 October 2020, has reported >17000 cases with >400 deaths.² Malawi confirmed its first three cases on 3 April and has reported >7000 cases and ~200 deaths over this same period.²

Methods

R2.2: “The setting section reads... ‘Lilongwe, is one of four regional referral hospitals in Malawi and the NNU has 75 cots. In contrast to SMCH, care in the NNU is mostly nurse-led.’

This appears incomplete. Could the authors clarify if they are contrasting overall care at the SMCH with care at the SMCH NNU or that of KCH as well.”

Thank you for spotting this ambiguity. We were specifically contrasting care at the neonatal unit of Sally Mugabe Central Hospital with that of the neonatal unit at Kamuzu Central Hospital. Care at the neonatal unit of Sally Mugabe Central Hospital, Zimbabwe, is predominantly doctor led, while care at the neonatal unit of Kamuzu Central Hospital, Malawi, is predominantly nurse led. We have amended this section for clarity.

Page 8, lines 12-24 now read (added text in blue italics):

SMCH is a public referral hospital in Harare, Zimbabwe. It has the largest of three tertiary NNUs nationwide with 100 cots. KCH, Lilongwe, is one of four regional referral hospitals in Malawi and the NNU has 75 cots. Neonatal care at SMCH is predominantly doctor led while neonatal care at KCH is mostly nurse led. Both units accept local and national referrals for specialist surgical care.

R2.3: “Although I am not a statistician, I have my reservations about the appropriateness of the choice of study design for this research question. The Interrupted time series analysis is a method of statistical analysis involving tracking a long-term period before and after a point of intervention to assess the intervention's effects. The time series refers to the data over the period, while the interruption is the intervention, which is a controlled external influence or set of influences. In this regard:

a. The date of first COVID-19 diagnosis, while a key event, cannot be considered as an intervention per se as it is not controlled. It is not the date of diagnosis but the government and health system response to this event (a lockdown) that is the ‘controlled intervention’ that has both direct and indirect impact.”

We generally agree with Dr Okomo’s concise description of the interrupted time series study design. However, we disagree that the date of the first cases of COVID-19 in a respective country cannot be considered an intervention (or interruption) for the purposes of a valid interrupted time series analysis. Interrupted time series is a quasi-experimental design; however, its statistical assumptions do not refer to the nature of the event(s) considered as the intervention. Hence, we argue it is not the ‘controlled’ nature of an intervention that determines its appropriateness for interrupted time series analysis, rather, it is knowing the exact date the intervention occurred that is important for interpretation of the results. Using the date of the first cases of COVID-19 allows us to clearly divide the time series into a pre-intervention period (before the first cases of COVID-19) and post-intervention period (after the first cases of COVID-19) in each country. We could have chosen to specifically evaluate the effects of the government response to the first cases of COVID-19 (i.e. a national lockdown). However, we chose not to restrict the analysis in this way, but instead evaluate the effects of the mere presence of COVID-19 within the country (be those effects related to governments’ response, individuals’ fears of attending health facilities, or other causes).

Furthermore, we refer to three examples in the literature where the first confirmed case of a disease has been used as the intervention of an interrupted time series analysis. These examples all relate to the 2014-2016 Ebola virus disease outbreak. Delamou et al. evaluated the impacts of the Ebola virus disease outbreak on maternal and child health services in Guinea, using the dates of the first and last cases of Ebola virus disease in the study region to define the ‘before’, ‘during’ and ‘after’ periods of their interrupted time series models.¹ Similarly, Magassouba et al. examined the impact of the Ebola

virus disease outbreak on tuberculosis surveillance activities in Guinea, again using the dates of the first and last cases of Ebola virus disease to define the study periods of an interrupted time series analysis.² Molinari et al. examined the impact of the first case of Ebola virus disease in the United States on the number of emergency department visits in Dallas-Fort Worth, Texas, using an interrupted time series design.³ As a final, COVID-19-related example, Mulholland et al. evaluated the impact of COVID-19 on emergency department attendances, and emergency and planned hospital admissions in Scotland using an interrupted time-series analysis with one change-point defined as the date the WHO declared a pandemic (11 March 2020), which we believe is analogous to a government reporting the first confirmed case of COVID-19.⁴

For these reasons, we believe that using the first confirmed cases of COVID-19 in Zimbabwe and Malawi as the intervention for an interrupted time series analysis is statistically valid.

R2.4: “The authors failed to mention if there was any national or regional lockdown in either country or the duration of such a lockdown.”

Thank you for pointing out this oversight. In Zimbabwe, borders closed to non-essential travel (excluding returning residents and cargo) on 23rd March 2020 and a full national lockdown was imposed on 30th March 2020, initially for a period of 21 days. This lockdown required people to stay at home except to buy basic food, fuel, or medicine supplies; to attend work for an essential service; or to obtain medical assistance. Non-essential businesses, restaurants, educational institutions, and non-essential transport services were also closed.⁵ This period of lockdown was subsequently extended to 11th June 2020, followed by phased relaxations of the restrictions until 5th January 2021, when a full lockdown was reinstated.

In Malawi, the Government announced on 20th March 2020 that all public events were banned, public gatherings were restricted to less than 100 people, and all educational institutions would close. On 1st April 2020, borders closed to non-essential travel (excluding returning residents and cargo). A full lockdown was announced to last for 21 days from 18th April to 9th May 2020. However, an injunction was issued by the High Court against this lockdown on 18th April 2020 and the lockdown was suspended.⁶ A further partial lockdown was announced on 9th August 2020, mandating the wearing of face masks in public spaces, closing places of worship, restaurants, and bars, and restricting public gatherings to less than 10 people. After public condemnation, these restrictions were revised several days later, reallowing gatherings up to 100 people.⁷

We have included these details in the ‘Setting’ section of the Methods:

Page 8, line 25 onwards now reads (added text in blue italics):

In response to the COVID-19 pandemic, Zimbabwe and Malawi have both implemented response measures in an attempt to control the outbreak. In Zimbabwe, the Government closed borders to non-essential travel within days of the first in-country confirmed case of COVID-19 and imposed a full national lockdown that lasted from 30th March 2020 to 11th June 2020, which was followed by phased relaxations of the restrictions.¹⁸ In Malawi, public events were banned and public gatherings restricted to fewer than 100 people on 20th March 2020, with all educational institutions closed several days later.¹⁹ Borders were closed to non-essential travel on 1st April 2020 and a full national lockdown was announced to last for 21 days from 18th April 2020; however, a High Court injunction prevented this. Further restrictions were announced on 9th August 2020, mandating the wearing of face masks in public, closing places of worship, restaurants, and bars, and restricting public gatherings to less than 10 people initially, although these were revised within days to reallow gatherings up to 100 people.²⁰ Schools in Malawi reopened on 7th September 2020.¹⁹

We have also updated Figure 5 ('Possible factors influencing the decrease in admissions to the neonatal unit') to include a node for 'National lockdowns'.

R2.5: "b. A bit more information on the setting in each NNU is necessary to provide context especially given the limitations intrinsic to interrupted time series analysis, one of which the author have mentioned to be the possibility that another event occurred close to the first case of COVID-19 in either country causing spurious observations. What is the usual pattern of admissions to each unit over the preceding few years? Is there usually any seasonal variation in deliveries at each hospital and NNU admissions? (In some studies, neonatal ward admissions have been shown to follow a seasonal pattern consistently peaking at certain months of the year. One needs at least 2 years data to establish seasonality). It is therefore important when considering any changes to the pattern of admissions that the authors compare events over the same times of the year before and after the intervention. If the first case of COVID-19 in either (or both) countries occurred about the same time that there is usually a drop in NNU admissions, then the authors cannot simply conclude their observations are due to the COVID-19 and would need to adjust for any seasonal trends. More appropriate would be the additional or excess decline in admissions above what would be expected for the same time of the year."

Robust data for the usual pattern of admissions to each neonatal unit prior to our analysis are difficult to obtain, which was one key driver for introducing the NeoTree app (Sally Mugabe Central Hospital in November 2018; Kamuzu Central Hospital in April 2019). We were able to obtain monthly admission numbers for Sally Mugabe Central Hospital and Kamuzu Central Hospital for 2017-2018 from registers kept in the records departments. In the following figures, we have plotted the number of monthly admissions to the neonatal unit for the years 2017-2018 alongside the monthly admission numbers from the NeoTree data for 2019-2020 (as used in our main manuscript analysis). Data for 2017-2018 should be interpreted cautiously as these rely on paper records; however, there is no obvious seasonality evident in this crude analysis and monthly admission numbers are generally variable from year to year.

Figure 1. Monthly admissions to the neonatal unit at Sally Mugabe Central Hospital, Zimbabwe, 2017-2020

Figure 2 Monthly admissions to the neonatal unit at Kamuzu Central Hospital, Malawi, 2017-2020

We include the above figures here for your interest, but do not present them in our revised manuscript due to the inherent limitations of comparing the paper records to the NeoTree data. However, we have added the following statement to our 'Limitations' section of the Discussion:

Page 18, line 29 onwards now reads (added text in blue italics):

Finally, the presence of seasonality is an important consideration in time series analyses. Unfortunately, prior to 2019, robust data for our outcomes are not available at either hospital due to a reliance on paper records, which could be lost or destroyed. Therefore, we could not adequately analyse seasonal patterns. However, for some outcomes, the scatterplots presented in our paper demonstrate a sudden shift in the trend at a defined time point in the series (around the first confirmed cases of COVID-19 or around time points coinciding with periods of industrial action). As similarly pronounced changes are not seen at other time points in the series, this would indicate the impact of the intervention despite any potential underlying seasonality.

References

1. Delamou A, Ayadi AME, Sidibe S, et al. Effect of Ebola virus disease on maternal and child health services in Guinea: a retrospective observational cohort study. *Lancet Glob Health* 2017;5(4):e448-e57. doi: 10.1016/S2214-109X(17)30078-5
2. Magassouba AS, Diallo BD, Camara LM, et al. Impact of the Ebola virus disease outbreak (2014–2016) on tuberculosis surveillance activities by Guinea’s National Tuberculosis Control Program: a time series analysis. *BMC Public Health* 2020;20(1):1200. doi: 10.1186/s12889-020-09230-2
3. Molinari NM, LeBlanc TT, Stephens W. The Impact of a Case of Ebola Virus Disease on Emergency Department Visits in Metropolitan Dallas-Fort Worth, TX, July, 2013-July, 2015: An Interrupted Time Series Analysis. *PLoS Curr* 2018;10 doi: 10.1371/currents.outbreaks.e62bdea371ef5454d56f71fe217aead0 [published Online First: 2018/04/07]
4. Mulholland RH, Wood R, Stagg HR, et al. Impact of COVID-19 on accident and emergency attendances and emergency and planned hospital admissions in Scotland: an interrupted time-series analysis. *J R Soc Med* 2020;113(11):444-53. doi: 10.1177/0141076820962447 [published Online First: 2020/10/06]
5. Government of Zimbabwe. Public Health (COVID-19 Prevention, Containment and Treatment) (National Lockdown) Order, 2020 Zimbabwe: Veritas Zimbabwe; 2020 [updated 28 Mar 2020; cited 2021 22 Jun]. Available from: <https://www.veritaszim.net/node/4046>.
6. Mzumara GW, Chawani M, Sakala M, et al. The health policy response to COVID-19 in Malawi. *BMJ Global Health* 2021;6(5):e006035. doi: 10.1136/bmjgh-2021-006035
7. Kaponda C. No COVID-19 lockdown still threatens livelihoods and trade in Malawi [Blog]. London School of Economics; 2020 [updated 25 Sept 2020; cited 2021 22 Jun]. Available from: <https://blogs.lse.ac.uk/africaatlse/2020/09/25/no-covid19-lockdown-threatens-livelihoods-trade-trust-malawi/>.

VERSION 2 – REVIEW

REVIEWER	Okomo, Uduak London School of Hygiene & Tropical Medicine, MRC Unit The Gambia
REVIEW RETURNED	21-Jul-2021

GENERAL COMMENTS	Thank you for the opportunity to review the revised manuscript I appreciate your efforts to address the comments raised. I however, still have several comments. Minor comment Introduction 1. The paragraph on NeoTree appears out of place in the introduction as it is not linked in any way to the study hypothesis and aim. It might be better placed in the methods Major Comments The major challenge with interpretation of the study finding is the absence of historical data on the pattern of admissions and mortality in the neonatal units studied. As I mentioned in my previous review, it is important when considering any changes to the pattern of admissions that the authors compare events over the same times of
---

	the year before and after the intervention. Without this, one cannot simply conclude the observed findings are due to the COVID-19 Discussion There are several discussion threads that have not been adequately explored in the context of the study findings  1. Although they have cited studies which reported increased rates of preterm birth in pregnant women with COVID, the authors have not explored the reasons for the observed decrease in gestational age and birthweight among neonates admitted to KCH. How many (if any) neonates were born to women with COVID in the study sites? 2. What do the authors think might be responsible for the observed increase in numbers of emergency caesarean section? Is mode of delivery and indication for C/S collected in the neo app? 3. The authors mentioned that neonates referred from outside KCH are more likely to be from lower-risk pregnancies that delivered in a health centre with higher gestational ages and birth weights. Why would low risk pregnancies be referred to KCH given that care is nurse-led? The authors also state that “further analysis should stratify by source of admission referral to clarify this finding but is supported by the fact that referrals were rigorously triaged by the on-call paediatrician during the pandemic, and that referrals from some areas were diverted away from KCH”. What were the reasons for the diversion of referrals away from some areas? Was the incidence of COVID higher in these areas or were the mothers suspected to have COVID and KCH was not equipped to handle this? 4. The authors state that the absence of increased rates of NE should be interpreted with caution as some neonates with NE may not have presented to a health facility – this would imply that they were born at home and either not recognised as having NE or not taken to the hospital for whatever reason (including the lockdown due to COVID). Is there any local/national data suggesting a corresponding increase in the number of home deliveries during this period, given the observed decrease in referrals and decrease in admissions? 5. The authors have suggested that the increase in mortality at KCH may be due to decreased GA and BW, and reduced rota of staff. This is all speculative given that comparable earlier data for the same time of the year is lacking.
--	---

VERSION 2 – AUTHOR RESPONSE

Reviewer 2: Dr Uduak Okomo

Minor comment
Introduction

R2.1: “The paragraph on Neotree appears out of place in the introduction as it is not linked in any way to the study hypothesis and aim. It might be better placed in the methods”.

We have moved this paragraph to the Methods section under the 'Data collection' subheading.

Page 8, line 19 onwards now reads (added text in blue italics):

"Data were collected prospectively using the Neotree application (app), an Android tablet-based quality improvement platform that aims to reduce neonatal mortality in low-resource settings.¹⁵ Developed in collaboration with local stakeholders, it is embedded in routine practice at two NNUs in Zimbabwe and Malawi, providing real-time clinical decision support, neonatal care education, and digital data capture.^{19 20}"

Major comments

R2.2: "The major challenge with interpretation of the study finding is the absence of historical data on the pattern of admissions and mortality in the neonatal units studied. As I mentioned in my previous review, it is important when considering any changes to the pattern of admissions that the authors compare events over the same times of the year before and after the intervention. Without this, one cannot simply conclude the observed findings are due to the COVID-19".

It is true that historical data on trends in our study outcomes would strengthen our findings; however, our interrupted time series models account for pre-COVID-19 trends within the paper's scope and the data available. For example, there could have been a significant increase in the number of admissions in the pre-COVID-19 year, but we do not have any data for that period before June 2019 and we could not address the effect from such historical data in our models. We have recognised and explained this limitation in the Discussion section.

There are few studies on the impacts of COVID-19 and the pandemic in low-resource settings, and our study was designed and implemented at a time where understanding of the disease was rapidly evolving. As stated in the first paragraph of our discussion (page 16, lines 36-40): "Although this study is descriptive, we can speculate about explanations for our results based on existing literature and discussions with local health workers." We understand the challenges and limitations of study design and data collection in a low-resource setting during an evolving global pandemic and, thus, our study is explicitly descriptive rather than explanatory. We hope that authors of future studies in similar low-resource settings will be able to compare their observations with ours to expand the evidence base for better causal inference.

Discussion

R2.3: "1. Although they have cited studies which reported increased rates of preterm birth in pregnant women with COVID, the authors have not explored the reasons for the observed decrease in gestational age and birthweight among neonates admitted to KCH. How many (if any) neonates were born to women with COVID in the study sites?"

The main aim of our study was to present our observations, but we have hypothesised potential reasons for the observed decrease in gestational age and birthweight among neonates admitted to KCH, but not SMCH. In our 'Discussion – Interpretation' section, we write "Studies have reported increased rates of preterm birth in pregnant women with COVID-19 compared to those without the disease, mostly from medically-induced preterm birth; although none of these studies were conducted in LMICs" (page 67, lines 49-54). At the time of data collection (towards the onset of the pandemic), COVID-19 specific variables were in the process of being added to the Neotree app, so data on the number of neonates born to women with COVID-19 for this period are not available. However, an increase in rates of preterm birth due to maternal COVID-19 is one plausible explanation of the decrease in gestational age and birth weight. Later, we write "Preliminary analysis suggests rates of

emergency caesarean section increased at SMCH and KCH, with a more marked increase at KCH (Appendix 6). This is one potential explanation for our findings” (page 67, lines 56-59). We suggest that iatrogenic preterm birth due to emergency caesarean section is another plausible explanation and, given the more marked increase in emergency caesarean section rates seen at KCH compared to SMCH, this might explain the observed differences in trends of gestational ages and birthweights between the two sites. As we state in the paper, this is speculative given the paucity of available data.

R2.4: “2. What to the authors think might be responsible for the observed increase in numbers of emergency caesarean section? Is mode of delivery and indication for C/S collected in the neo app?”

This is unclear. When we found a decrease in gestational age and birth weight of neonates at KCH, we decided to explore trends in mode of delivery at each site but did not conduct robust analysis on these data, as this was beyond the intended scope of our study. With each finding, there are many potential avenues that could be further explored in more detail, but this is simply not possible within one study. However, for yours and readers’ interest, we present the trends in commonest reasons for caesarean section in the figures below and have added these figures to Appendix 6 of our supplementary material.

Previous caesarean section scar is the commonest indication for elective caesarean section at both SMCH and KCH, but we see no marked trends in the pre-COVID-19 vs. post-COVID-19 periods from visual inspection of the data (Figure 1). Pre-eclampsia, eclampsia and fetal distress are the commonest indications for emergency caesarean section at SMCH and KCH and, again, there were no obvious trends apparent when comparing the pre-COVID-19 to post-COVID-19 periods (Figure 2). There does, however, appear to be an increase in the relative proportion of emergency caesarean sections due to eclampsia compared to pre-eclampsia at SMCH during the period of industrial action by doctors, but this is speculative without further robust analysis.

Figure 1: Trend in reason for elective caesarean section per week

Figure 2: Trend in reason for emergency caesarean section per week

R2.5: “3. The authors mentioned that neonates referred from outside KCH are more likely to be from lower-risk pregnancies that delivered in a health centre with higher gestational ages and birth weights. Why would low risk pregnancies be referred to KCH given that care is nurse-led?”

To answer this, it may be helpful to explain in more detail what we mean by ‘nurse-led’ in this context. At KCH, care of sick neonates is nurse-led in so much as the nurses will admit and discharge all babies, instigate treatment, and carry out all basic neonatal care (including feeding, thermoregulation, and monitoring). However, all critically ill neonates are also reviewed urgently by a clinical officer or more senior clinician as soon as possible, and daily ward rounds are carried out by a clinician depending on staff availability. By comparison, in primary health centres, there are no doctors or clinical officers and not all centres have basic equipment (e.g. methods for oxygen delivery). Therefore, if a baby is identified as being sick or vulnerable and deemed likely to benefit from escalation of care, they will be referred to KCH or one of the other three central, tertiary care units in Malawi. There is only one qualified neonatologist in Malawi for a population of 19.13 million people, and a handful of paediatricians with a special interest in neonatology. Hence, in general, neonatal care is ‘nurse-led’ across all Malawian health facilities.

R2.6: “The authors also state that “further analysis should stratify by source of admission referral to clarify this finding but is supported by the fact that referrals were rigorously triaged by the on-call paediatrician during the pandemic, and that referrals from some areas were diverted away from KCH”. What were the reasons for the diversion of referrals away from some areas? Was the incidence of COVID higher in these areas or were the mothers suspected to have COVID and KCH was not equipped to handle this?”

Before the pandemic, direct referrals from primary health centres to tertiary care facilities (e.g. KCH) were sometimes accepted. However, during the pandemic, the correct referral route (from primary health centre to secondary referral centre, and then to KCH) was more strictly adhered to, provided it was clinically appropriate. The on-call paediatrician ensured that direct referrals from primary health centres for neonates that could be managed in a secondary referral centre were ‘diverted’ away from KCH. As far as we are aware, the incidence of COVID-19 was not higher in these areas. KCH was equipped to handle mothers suspected of COVID-19.

For clarity, we have amended the relevant section of our Discussion.

Page 17, line 36 onwards now reads (added text in blue italics):

“Further analysis should stratify by source of admission referral to clarify this finding, but the relative reduction in outside referrals is supported by the fact that referrals were rigorously triaged by the on-call paediatrician during the pandemic, and that referrals from some areas were diverted away from KCH to more appropriate centres for the level of care required.”

R2.7: “4. The authors state that the absence of increased rates of NE should be interpreted with caution as some neonates with NE may not have presented to a health facility – this would imply that they were born at home and either not recognised as having NE or not taken to the hospital for whatever reason (including the lockdown due to COVID). Is there any local/national data suggesting a corresponding increase in the number of home deliveries during this period, given the observed decrease in referrals and decrease in admissions?”

A search of PubMed in December 2021 using the search terms “(Zimbabwe OR Malawi) AND COVID-19[Mesh] AND Home Childbirth[Mesh]”, or appropriate synonyms, returned no results. However, a recent mixed methods study in Nampula, Mozambique’s third largest city, suggested that hospital deliveries fell by 4% ($p = 0.046$) and home deliveries increased by 74% ($p = 0.074$).¹

R2.8: “The authors have suggested that the increase in mortality at KCH may be due to decreased GA and BW, and reduced rota of staff. This is all speculative given that comparable earlier data for the same time of the year is lacking.”

See our response to R2.2.

Statistical advisor

SA.1: “I can see the point of the reviewer. In Zimbabwe, the first case was 20th March and lockdown was 30th March to 11th June, followed by phased restrictions. In Malawi, the first case was 3rd April, and there was some form of restrictions from 20th March to 7th September.

“So the two analyses differ in that in Zimbabwe the point of the intervention was before lockdown,

where as in Malawi it was during lockdown.

“It is really the definition that of the impact of lockdown or the fact that COVID was in the country. This presents the latter. I think this is ok.”

Thank you for your careful consideration of this point. The first cases of COVID-19 in each country (i.e. the presence of COVID-19 in the country, as you say) defines our intervention and our hypothesised reason for the change in trend for each outcome.

SA.2: “My main concern though is the lack of discussion about the strikes (with dates) – I couldn’t see dates of these and what services were effected. This seems to have had big effects esp. for admissions and seems more interesting than ‘the first case’ as the interruption.”

The primary intent of our study was not to explore the effects of industrial action in Zimbabwe, but we agree this is an interesting aspect to mention in more detail. By way of background, we have added the following paragraph to the ‘Methods, Setting’ subsection of our manuscript.

Added text in blue italics:

“Industrial action by health workers in Zimbabwe

“Two periods of national industrial action occurred in Zimbabwe during our study. Doctors went on strike from 3 September 2019 to 22 January 2020 (pre-COVID-19 period) citing insufficient pay and poor working conditions, which put significant pressure on the public health system.¹⁹ Additionally, there was a period of strikes by nurses from 17 June to 9 September 2020 (post-COVID-19 period) over pay and availability of personal protective equipment during the pandemic.²⁰”

Effects related to the nurses’ strike in the post-COVID-19 period are already detailed in the ‘Results’ section and discussed further in the ‘Discussion’. For example:

Page 63, lines 31-36 reads:

“However, this model estimated that admissions fell by 48% during the nurses’ strike period (RR 0.52, 95%CI 0.41-0.66; $p < 0.001$)...”

We now report pertinent effects related to the doctors’ strike in the relevant ‘Results’ section for each outcome, as follows:

Added text in blue italics:

“Outcome 1: Admissions to the neonatal unit

“... and by 51% during the pre-COVID-19 doctors’ strikes (RR 0.49, 95%CI 0.41-0.60; $p < 0.001$).”

“Outcome 3: Source of admission referral

“... However, this model did imply a 39% relative increase in the percentage of outside referrals during the doctors’ strikes in the pre-COVID-19 period (RR 1.39; 95%CI 1.20-1.61; $p < 0.001$).”

SA.3: “There should be adjustment for seasonality here. Data is only collected 6-7 months after the ‘first covid case’, albeit weekly”.

We have dealt with weekly seasonal patterns by using weekly data windows in our interrupted time series models, and by fitting seven-day moving average smoothers for visual and exploratory analyses. As you suggest, there may be yearly or semestral patterns which should be accounted for even after this deseasonalisation. To adjust for seasonality, we modelled seasonal patterns as cosine

functions with variable amplitude and shift. We tested models fitting a cosine function with 6-month period, a cosine function with 12-month period, and with both harmonic terms. Models were selected by minimising the Bayesian Information Criterion (BIC) and by comparing goodness-of-fit with the χ^2 -test for nested models. Adjusting for seasonality, as mentioned in the paper, did not improve the fit of any of the models tested and, thus, all presented models are unadjusted for seasonality. We present unadjusted and seasonally-adjusted models in Appendix 5 of our supplementary material. Incidentally, we remodelled count data using generalised linear models with Poisson or negative binomial distributions with logarithmic link functions, instead of quasi-Poisson models as in our original submission.

References

1. das Neves Martins Pires PH, Macaringue C, Abdirazak A, et al. Covid-19 pandemic impact on maternal and child health services access in Nampula, Mozambique: a mixed methods research. *BMC Health Serv Res* 2021;21(1):860. doi: 10.1186/s12913-021-06878-3